evolution, genetics

ageing, damage accumulation, disposable soma, developmental theory of ageing, life-history theory, senescence

**Authors for correspondence:**
Alexei A. Maklakov
e-mail: a.maklakov@uea.ac.uk
Tracey Chapman
e-mail: t.chapman@uea.ac.uk

# Evolution of ageing as a tangle of trade-offs: energy versus function

Alexei A. Maklakov and Tracey Chapman

School of Biological Sciences, University of East Anglia, Norwich Research Park, Norwich NR4 7TJ, UK

(iD) AAM, 0000-0002-5809-1203; TC, 0000-0002-2401-8120

Despite tremendous progress in recent years, our understanding of the evolution of ageing is still incomplete. A dominant paradigm maintains that ageing evolves due to the competing energy demands of reproduction and somatic maintenance leading to slow accumulation of unrepaired cellular damage with age. However, the centrality of energy trade-offs in ageing has been increasingly challenged as studies in different organisms have uncoupled the trade-off between reproduction and longevity. An emerging theory is that ageing instead is caused by biological processes that are optimized for early-life function but become harmful when they continue to run-on unabated in late life. This idea builds on the realization that early-life regulation of gene expression can break down in late life because natural selection is too weak to optimize it. Empirical evidence increasingly supports the hypothesis that suboptimal gene expression in adulthood can result in physiological malfunction leading to organismal senescence. We argue that the current state of the art in the study of ageing contradicts the widely held view that energy trade-offs between growth, reproduction, and longevity are the universal underpinning of senescence. Future research should focus on understanding the relative contribution of energy and function trade-offs to the evolution and expression of ageing.

## 1. Introduction

> It is indeed remarkable that after a seemingly miraculous feat of morphogenesis a complex metazoan should be unable to perform the much simpler task of merely maintaining what is already formed.
>
> —George C. Williams

Ageing, or senescence, is a physiological deterioration of an organism with advancing age, which reduces reproductive performance and increases the probability of death [1,2]. Despite the fact that ageing reduces Darwinian fitness, it is ubiquitous and represents an integral part of the life course of most species on Earth [2,3]. It was originally believed that ageing is restricted only to humans, captive animals, and livestock because animals in nature die from predation, competition, and parasites before they senesce. Therefore, ageing was predicted to lie largely outside the realm of natural selection. However, this view has been overturned in recent years by a string of outstanding studies in natural populations that have definitively demonstrated that ageing also occurs in the wild and is very common (reviewed in [3]). Nevertheless, there is a remarkable diversity in the patterns of ageing across the tree of life, with some species showing negligible rates of senescence in either age-specific reproduction, mortality, or both [4]. To explain this variation, evolutionary biologists and biogerontologists have sought to understand why ageing evolves, what determines variation in lifespan and rates of ageing, what are the proximal causes of ageing, and are they evolutionarily conserved? Such an understanding requires an integrated approach in which evolutionary concepts are used to guide the research into the mechanisms of ageing, while knowledge of the mechanisms is then used to support or reject different evolutionary theories.

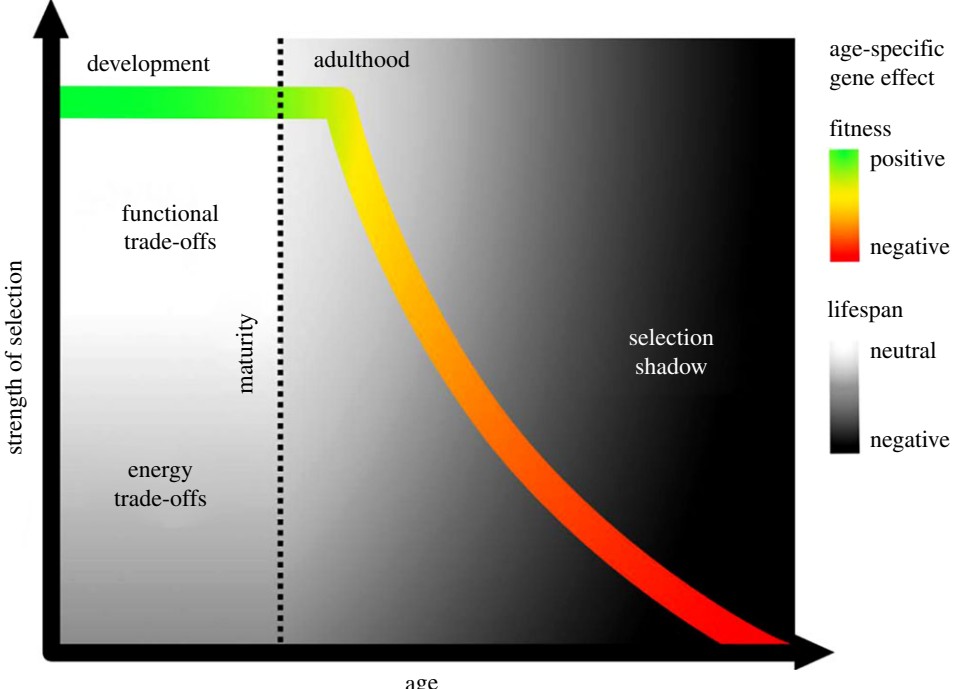

**Figure 1.** The strength of age-specific selection is maximized during pre-reproductive development but declines after sexual maturation with advancing adult age and reaches zero at the age of last reproduction [11–13,15]. The colours along the selection gradient line represent the effect of an antagonistically pleiotropic (AP) allele on fitness across the life course, from positive green early in life to strongly negative red in late life. The shading of the background represents the effect of an AP allele on lifespan across the life course, from neutral white to strongly negative black. The classic AP allele, as envisioned by Williams [10], will have a positive effect on fitness during development but a negative effect on fitness in late life. However, the effects of such an AP allele on lifespan will vary across the life course depending on whether the trade-off between lifespan and other fitness-related traits is based on energy or function. The negative effect on lifespan can result from competitive energy allocation between development, growth, and reproduction on the one hand, and somatic maintenance on the other hand, resulting in energy trade-offs as suggested by the 'disposable soma' theory [16]. Under energy trade-offs, damage accumulation due to insufficient repair starts early in life and accumulates through the ages until the demise of an organism and lifespan extension is always costly. However, functional trade-offs result from suboptimal regulation of gene expression in late life resulting in suboptimal physiological function. Under functional trade-offs, optimizing gene expression in adulthood improves both fitness and lifespan, without developmental costs. (Online version in colour.)

## 2. Why do organisms age?

This key and enduring question requires qualification: are we asking the precise reasons why you or I might be ageing, or alternatively why ageing evolved in the first place? The failure to clearly distinguish between these proximate and ultimate questions, and why it even matters has, arguably, resulted in a fair amount of confusion within the broader study of ageing. The aim of this review is to marry these approaches and to facilitate a more complete understanding of the biology of ageing by integrating the latest mechanistic advances with the evolutionary theory. Through this, we will promote the idea that these approaches are complementary, synergistic, and can help in the development of age-specific life-history theory.

Evolutionary theories of ageing have been extensively reviewed [1,2,5,6] and are only briefly summarized here. These theories rely on the axiom that selection maximizes fitness, not necessarily lifespan. Therefore, ageing is associated with selective processes to build vehicles for successful reproduction [7,8]. The key idea underpinning the evolutionary theory of ageing is that the strength of natural selection on a trait declines after sexual maturation and with advancing age [9–13] resulting in Haldane's [14] famous 'selection shadow' (figure 1). This is because non-ageing-related extrinsic mortality reduces the probability of late-life reproduction and old individuals in the population have already produced a large part of their lifetime reproduction and passed on their

genes resulting in a decline in selection gradients on mortality and fertility [11–13,17]. This single, fundamentally important insight led to the formulation of the major theories of ageing.

*Mutation accumulation*: in this, mutations with late-life effects can accumulate and be transmitted through the germ line [9]. Ageing here occurs due to the summation of randomly acquired deleterious effects that are manifested only late in life [18]. Following from the premise of the 'selection shadow', ageing at late ages has relatively little impact upon an organism's overall fitness. Early formulations of MA assumed narrow 'windows' for mutational effects during the life course of the organism, and models based on such assumptions predicted a rapid increase in mortality after the end of reproduction, or 'walls of death' that are rarely seen in nature. Subsequent models considered the possibility for positive mutational effects across adjacent age-classes and explored the extent to which MA could occur even if genes ultimately responsible for ageing had mildly deleterious effects in early life [6,19,20]. These models allowed for post-reproductive lifespan, a more gradual increase in mortality rate with age and a decline in mortality rates at a very late age. MA theory makes no specific assumptions about which types of pathways should underpin ageing, as the accumulation of mutational effects could in theory occur across random loci. MA has some support (reviewed in [21]), though recent discussions have

highlighted that it may not be consistent with the discovery of the molecular signalling pathways that potentially underpin ageing across many animal groups and appear to be evolutionarily conserved [21–23].

*Antagonistic pleiotropy*: is the evolutionary theory of ageing which recognizes that genes often have multiple or pleiotropic effects and that a beneficial effect of a gene early in an organism's life can be strongly selected even if that gene causes a negative effect later in life [10]. Because selection gradients on survival and fertility decline with age, early-life beneficial effects are likely to be strongly positively selected, and deleterious late-life effects can persist because selection is weak and cannot eliminate them. Antagonistic pleiotropy (AP) emphasizes the importance and inevitability of trade-offs between different life-history traits across early and late life. To ascertain whether MA or AP was the dominant paradigm, many studies have examined whether enhanced success in reproduction early in life is inevitably associated with decreased lifespan or increased ageing. Laboratory evolution experiments have successfully selected for increased late-life fitness and observed decreased early-life fitness as a correlated response [24,25] in line with AP theory. Others selected directly for increased survival and observed decreased reproductive output [26]. The identification of individual alleles with AP effects has also strengthened support [27–29]. As noted above, Williams originally suggested the types of loci that could show antagonistic effects and, while at first an abstract concept, the finding of genes with the appropriate profile of antagonistic effects provides intriguing support for AP. One example is found in the sword tail fish (*Xiphophorus cortezi*), in which individuals that carry the dominant *Xmrk* oncogene simultaneously have increased the risk of melanoma and a selective size advantage [30].

# 3. Energy trade-offs between growth, reproduction, and longevity

While the logic of the AP theory of ageing [10] is straightforward, supported by mathematical modelling [11] and quantitative and molecular genetics [2,27], it does not explain which physiological processes actually result in organismal senescence. Connecting evolutionary and mechanistic explanations for ageing is important because (i) this knowledge builds a general understanding of the ageing process, (ii) knowing which physiological processes contribute to organismal senescence could provide powerful tests of ultimate ageing theory. Perhaps the most accomplished physiological/mechanistic account of AP to date is the 'disposable soma' theory of ageing (DST) [7,16,31]. While this model was developed as an independent evolutionary theory of ageing, and is sometimes presented as such in the literature alongside MA and AP, we agree with many researchers in the fields of evolutionary biology, ecology, and biogerontology that DST represents a physiological explanation of AP.

The premise of DST is that most organisms develop in environments in which resources are limited at least during some part of their lives. Because growth, reproduction, and somatic maintenance require energy, it is reasonable to expect that limited resources will be allocated between these different traits to maximize fitness. These are the energy trade-offs that underlie the DST and more broadly

life-history theory itself [32]. Cellular damage occurs constantly and can result from direct damage to the genome and from accumulation of insoluble protein compounds that interfere with cell function. While organisms possess many maintenance and repair mechanisms that can be deployed for genome repair as well as to re-fold or clear away misfolded proteins, it may ultimately be beneficial to invest in such maintenance and repair only to maximize the organismal function during the expected period of life, which will be determined by environmental mortality risk [31]. There is no benefit of investing in high fidelity and long-term maintenance and repair to produce an organism that shows negligible senescence, but which is highly likely to be quickly predated or killed by pathogens.

## (a) Increased reproduction accelerates ageing and vice versa

There is a wealth of evidence to support the existence of genetic trade-offs between early- and late-life fitness. Classic experimental evolution studies in *Drosophila* revealed negative early- versus late-life genetic correlations for fitness by selecting flies for early or late age at reproduction [15,33,34]. Follow-up studies in *Drosophila* and other invertebrates [25,35,36] using selection regimes that controlled for potentially confounding factors such as larval density also confirmed that enhanced late-life reproduction and survival was negatively correlated with early-life reproduction. Such studies are often cited in direct support of the DST [37]. However, while they provide evidence for a genetic correlation that is consistent with AP, they do not identify whether the underlying mechanisms are as predicted by the DST.

More direct support for the DST comes from the growing number of reports of trade-offs between investment in early-life performance and late-life performance in natural populations [38–41]. These 'ageing in the wild' studies have contributed three major advances: first, to help dispel the myth that ageing in nature is rare; second, to provide evidence that early–late life trade-offs shape individual life histories in natural populations; and third, to show that ageing in nature is plastic and depends strongly on the early-life environment.

Further evidence for DST comes from the experimental studies of natural populations. Field experiments with Collared Flycatchers (*Ficedula albicollis*) on the Swedish island of Gotland have shown that birds that reared an experimentally enlarged brood of nestlings laid smaller clutches later in life than did control birds [42,43]. Subsequent studies in other birds have shown that artificially increased brood size can also negatively affect survival [44–46]. Similarly, artificially increased litter sizes are associated with reduced survival in bank voles (*Clethrionomys glareolus*) [47]. Interestingly, in several mammalian studies, experimental increases to litter sizes did not result in reduced survival or reproduction of the parents, but instead reduced offspring size or survival [48–50]. These results suggest that there are significant costs of reproduction, and that animals can differentially allocate their investment between parental survival, offspring number, and offspring quality. These experimental studies come closest to linking increased reproduction with accelerated ageing, but do not yet identify the underlying mechanisms involved.

Hence, while numerous studies provide evidence for potential energy trade-offs in natural settings, the important

additional steps are: (i) to demonstrate energy reallocation (e.g. [51]), (ii) to identify mechanisms that contribute to these trade-offs in order to evaluate the relative importance of the DST in the evolution and expression of ageing.

## 4. Trade-offs between reproduction and survival can be uncoupled

Despite cross-taxonomic support for the idea that competitive energy allocation between growth, survival, and reproduction can contribute to senescence, the last two decades have seen an increase in a number of studies that challenge the centrality of energy trade-offs in ageing (reviewed in [15,52–56]). For example, reproduction increases metabolism, which leads to increased generation of reactive oxygen species (ROS) that can contribute to cellular damage and senescence. However, studies in fruit flies suggest that direct experimental reduction in ROS production via mitochondrial uncoupling proteins (UCPs) extends lifespan without a concomitant decrease in fecundity or physical activity [57]. Similarly, experimental downregulation of the nutrient-sensing target-of-rapamycin (TOR) signalling pathway in *Drosophila melanogaster* extends lifespan both in sterile flies (via rapamycin, [58]) and in fertile flies without negative effects on reproduction (via torin, [59]).

Some of the strongest empirical evidence against the energy allocation trade-offs being the universal cause of ageing comes from experimental studies that have directly uncoupled increased longevity from reduced fecundity. For example, downregulation of the evolutionarily conserved insulin/IGF-1 signalling (IIS) pathway that shapes development, growth, reproduction, and longevity increases lifespan but can also lead to arrested development and/or reduced early-life reproduction. However, a classic study by Dillin *et al*. [60] showed that the negative effects of reduced IIS on reproduction and positive effects on longevity can be uncoupled depending upon when during the life course of the organism the changes to IIS occur. Early-life downregulation of IIS signalling by RNA interference knockdown of *daf-2* gene expression in *Caenorhabditis elegans* starting at the egg stage or in early larval development extended lifespan, but resulted in reduced early-life fecundity. However, allowing the nematodes to develop normally and reach sexual maturity prior to IIS downregulation completely eliminated the negative effects of this manipulation on development and reproduction but prolonged lifespan to the same extent. This study definitively showed that while *daf-2* expression underpins negative genetic correlation between reproduction and survival, this correlation can be uncoupled by precise age-specific optimization of gene function. Essentially, wild-type levels of *daf-2* expression contribute to senescence and shorten the life of the worm not because of accumulation of unrepaired molecular damage starting from early life onwards, but because of the damage directly created during adulthood. Hence, optimizing gene expression in adulthood reduced damage and increased lifespan, without negative consequences for other life-history traits.

These findings prompt the question of what kind of damage might be created by suboptimal gene expression leading to suboptimal IIS function in adulthood. There is good evidence that misfolded protein aggregates that accumulate in cells with age contribute to cellular senescence, and several studies have linked reduced protein synthesis with increased longevity [61–65]. IIS signalling promotes protein synthesis [66,67] and, consequently, exceptionally long-lived *daf-2* *C. elegans* mutants whose IIS signalling is reduced exhibit a marked reduction in translation [68]. This suggests that superfluous protein synthesis in adulthood contributes to cellular senescence and organismal death. Nevertheless, protein synthesis is not the only anabolic process controlled by IIS signalling, and recent work in *C. elegans* has also linked superfluous yolk production in late life to senescence [69].

There are two objections to the idea that tinkering with age-specific gene expression can postpone ageing and increase lifespan without apparent fitness costs. First, it is possible that worms with reduced IIS signalling in adulthood are underperforming across different environments. Arguing against this, IIS mutants are known to be resistant to a wide range environmental stressors and exhibit increased tolerance to heat, cold, certain pathogens, to oxidative stress caused by visible light and radiation. These studies suggest that downregulation of IIS signalling in adulthood could improve the fitness of *C. elegans* nematodes across a range of ecologically relevant environments, although more research is necessary to fully test this assertion. The second objection is the potential for deleterious inter-generational effects of experimentally adjusted physiology. Most studies focus on the trade-off between longevity and offspring number, but increased investment into the parental soma could come at the cost of offspring quality. There are at least two different routes through which the putative trade-off between parental longevity and offspring quality can be realized [55]. First, increased reproduction can result in reduced parental investment in terms of quantity and/or quality of resources provided by the parents to the offspring. Second, increased investment into parental soma can be traded-off with investment into germline maintenance and repair resulting in increased number of de novo germline mutations in offspring. However, recent work has showed that the offspring of parents treated with *daf-2* RNAi during adulthood had higher reproduction, similar lifespan, and higher Darwinian fitness than their control counterparts [70]. Taken together, these studies suggest that wild-type IIS signalling in nematodes optimizes development at the cost of reduced survival in adulthood and reduced offspring fitness, making the wild-type *daf-2* an AP allele whose late-life cost does not result directly from energy trade-offs.

## 5. Function trade-offs: from Williams to the developmental theory of ageing

### (a) Origins of the theory

While damage accumulation resulting from energy trade-offs between reproduction and maintenance is generally viewed as the leading physiological/mechanistic explanation for the evolution and expression of ageing via AP, it is interesting that Williams himself used a very different example to illustrate the action of a putative AP allele [10]. He envisioned an allele with a beneficial effect on bone calcification in the developing organism, but that causes calcification of arteries in adulthood. Such an allele could have an overall positive effect on fitness and become established in the population. This is an example of a functional trade-off, where the same

physiological process is beneficial for fitness in early-life (e.g. during development), but detrimental for fitness in late life (e.g. post-sexual maturation). Williams suggested that selection could lead to the evolution of a modifier gene to suppress excessive calcification of arteries with age, but noted that such suppression is unlikely ever to be fully effective given weak late-life selection [10].

## (b) The developmental theory of ageing

Williams' ideas have been developed further in recent decades following the above logic, by explicitly linking the development of an organism to its senescence with advancing age [5,8,71–74]. In its broadest sense, the developmental theory of ageing (DTA) argues that ageing and longevity are shaped by the physiological processes that are optimized for early-life development, growth, and reproduction and are not sufficiently optimized for late-life function [8,71]. Importantly, there is a clear distinction between the classical damage accumulation paradigm as envisioned by the DST and the DTA. Damage accumulation hypotheses, including the DST, predict that increased investment in somatic maintenance will reduce cellular damage and increase longevity at the cost of reduced growth and reproduction. Because the organisms are predicted to optimize energy allocation between life-history traits to maximize fitness, experimental reallocation of energy should result in fitness costs. Contrary to this, the DTA predicts that it is possible to optimize age-specific gene expression to increase longevity without incurring costs to growth and reproduction, because longevity is curtailed by suboptimal physiology in adulthood rather than by the lack of resources for somatic maintenance. Hence, the DTA offers an explanation for the results of *daf-2* studies in *C. elegans*, which exemplify how modification of gene expression can have negative fitness consequences when applied during development but positive when applied during adulthood [60,70]. Consistent with this, an RNAi screen of 2700 genes involved in *C. elegans* development identified 64 different genes that are detrimental when deactivated during development, but which extend longevity when deactivated in adulthood [75,76]. It will be instrumental for our understanding of ageing to study the fitness consequences of age-specific optimization of gene expression across a broad array of physiological processes. The decline in selection gradients with age makes it logical that ageing evolves as a combined effect of many alleles with beneficial early-life and detrimental late-life effects, precisely because weak selection struggles to fully optimize gene expression in late life.

## (c) Excessive biosynthesis as a proximate cause of ageing

Recently, the concept that ageing results as a consequence of suboptimal gene function in later life has been mechanistically linked with the idea that superfluous nutrient-sensing signalling during adulthood can lead to excessive biosynthesis resulting in cellular hypertrophy, cellular senescence, and organismal senescence. The 'hyperfunction' idea provides a direct mechanistic mode-of-action hypothesis to explain how the continuation of a developmental programme can cause damage in late life and contribute to ageing [5,72,74,77]. In doing so, this hypothesis has moved the field forward by stimulating new studies aimed specifically at identifying the negative consequences of superfluous biosynthesis with advancing age. It has support from *in vitro* studies of cell cultures which suggest that, when cell proliferation is arrested, high levels of growth signalling can trigger the cells to transition from a quiescent to senescent state, while reduced growth signalling reduces cellular hypertrophy and senescence [77,78]. Cellular hyperfunction is predicted to result in organismal senescence and death, and demonstrating this link is vital for future tests of the 'hyperfunction' hypothesis. Most recently, Ezcurra *et al.* [69] showed that IIS signalling in nematodes promotes the conversion of gut biomass to yolk. While this process contributes to reproduction and is beneficial early in life, it is detrimental to the survival of ageing non-reproducing nematodes. Thus, one important factor that causes death in old worms is not an accumulation of unrepaired damage but the opposite: active yet costly conversion of gut biomass to unused yolk in the cells, resulting in intestinal atrophy and senescent obesity in old worms.

## (d) Age-specific trade-offs in optimal function

The 'hyperfunction' hypothesis is firmly rooted in the logic of age-specific trade-offs in gene function championed by Williams [10] and further developed by Hamilton [11]. This hypothesis provides a clear and detailed physiological explanation for how excessive biosynthesis in late life can contribute to cellular and organismal senescence by linking the quantitative genetic AP theory with the proximate DTA theory via nutrient-sensing molecular signalling within cells. Nevertheless, we believe that this concept can and should be broadened further, to encompass *all* possible ways in which suboptimal gene expression leading to suboptimal physiology in late life can contribute to the evolution and expression of ageing. For example, excessive biosynthesis may prove to be an important physiological mechanism of senescence, but there is no reason to expect it to operate across all taxa. Just as hyperfunction of nutrient-sensing signalling seems to play an important role in the age-related demise of nematodes, other physiological processes, some running 'too high' and some running 'too low', can contribute to ageing [8,71]. Moreover, in some cases, the developmental programme can actively downregulate certain physiological processes that would be beneficial in late life. For example, heat-shock resistance is actively downregulated in *C. elegans* nematodes upon sexual maturation, resulting in accumulation of insoluble protein compounds in the cells leading to disrupted proteostasis and, ultimately, senescence [79]. This 'hypofunction' of molecular chaperones has prompted researchers to suggest that sexual maturation marks the onset of ageing in *C. elegans*, much in line with Hamilton's [11] predictions. Understanding which physiological processes that shape senescence are more prevalent in different taxonomic groups, and why, should be the focus of research on the biology of ageing.

## 6. The hidden costs of longevity

The study of the evolutionary ecology of ageing has been driven by the search for energy trade-offs between life-history traits. Above, we have emphasized the role of non-energy-based trade-offs in the evolution of ageing (see also figure 2).

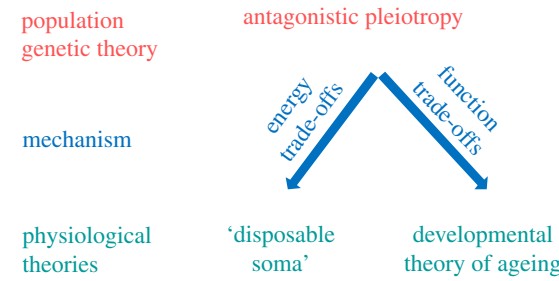

**Figure 2.** AP is a population genetic theory of ageing postulating that ageing evolves via alleles that have positive fitness effects early in life and negative fitness effects late in life [10]. Because the strength of age-specific selection is maximized early in life [11–13], such alleles can be beneficial and will be selected for. There are two proximate physiological mechanisms that account for AP: energy and function trade-offs between development, growth, reproduction, and survival. The DST [7,16,31] focuses on the energy trade-offs between growth, reproduction, and survival, while the DTA [8,71] focuses on gene expression optimized for development and early-life function. The relative importance of these two processes in the evolution of ageing is unknown, as most studies testing for age-specific fitness do not identify the physiological mechanisms. (Online version in colour.)

However, if energy-based trade-offs are not detectable in standard life-history assays, it may be fruitful to ask where the classic energy-based costs of longevity might sometimes be hidden and perhaps overlooked. First, perhaps the most obvious reason for the lack of fitness costs of lifespan extension is that fitness may often be assessed in one or two structurally simple environments. Laboratory studies using small, rapidly reproducing organisms provide a fertile ground for evaluation of the fitness consequences of genetic, environmental, or pharmacological lifespan extensions across a wide range of complex, ecologically relevant environments. There is untapped power in these systems to run controlled experiments in which animals encounter natural fluctuations of light, temperature, humidity, food and mate availability, presence of pathogens and predators across their life course, as well as across generations. While such experiments may be challenging and expensive, they are certainly feasible. Recent studies in related fields suggested that adding complexity to laboratory evolution can provide important novel insights into the evolutionary processes [80–82]. Second, fitness effects of lifespan extension are often assessed within one, and rarely two, generations. Measures of the quality of offspring, or perhaps grand-offspring, may represent an important and missing fitness component. The offspring of parents whose somatic performance has been artificially improved could experience a reduced quantity and/or quality of developmental resources, reduced parental care, or increased germline mutation rate. Such inter-generational trade-offs between parental longevity and offspring fitness have been demonstrated recently, but much more work is needed to establish whether such effects are common.

## 7. Integrated view of ageing: ultimate and proximate reasons

An integrated view, drawing together the proximate and ultimate concepts discussed above and building from the rapidly accumulating new knowledge is likely to revolutionize the study of ageing over the next few decades. We advocate

this approach to fully evaluate the relative importance of different ageing pathways, and to use these data to distinguish between different evolutionary hypotheses (figure 2). Both energy and function trade-offs are likely to contribute to the evolution and expression of ageing across organisms. Nevertheless, it is possible and necessary to establish their relative importance in the evolution of ageing across taxa and in shaping individual differences in age-related physiological decline. Understanding and quantifying the contribution of different types of trade-offs will not only help answer why do most organisms senesce as they grow old, but also will guide the efforts in the field of applied biogerontology.

Studies of ageing in laboratory model organisms, such as yeast, nematodes, fruit flies, and mice, have provided a wealth of mechanistic detail underpinning ageing-related traits such as lifespan, fecundity, and locomotion. However, they have much less often estimated fitness consequences of experimental manipulations. More studies explicitly linking gene function with age-specific fitness are required to understand whether old animals generally die from unrepaired damage that they slowly accumulated over their lives or from newly acquired damage resulting from a rogue developmental programme.

Studies of ageing in wild populations have often focused on correlations between measures of reproductive effort and survival, without exploring the underlying physiology. For example, high early-life reproductive performance is often associated with accelerated senescence in the wild, but no study as yet has causatively linked increased reproduction to damage accumulation to senescence. However, there is scope for experimental work with vertebrate species in natural and semi-natural environments. For example, dietary restriction-mimicking compounds that downregulate nutrient-sensing signalling and prolong lifespan in laboratory studies, such as rapamycin, can be administered at different ages via food and the effects on Darwinian fitness evaluated accordingly.

## 8. Conclusion

Trade-offs underlie the evolution of ageing. The two proximate theories focus on either imperfect repair due to competitive energy allocation (the DST) or imperfect function due to suboptimal gene expression after sexual maturation (the DTA). While both rely on the principle of AP, they make distinct predictions with respect to the relationship between growth, reproduction, and survival. We need to understand how trade-offs work in order to distinguish whether they are primarily energy-based or function-based. Distinguishing between these mechanisms may have profound practical consequences. For example, should DTA represent the dominant paradigm, it could significantly boost our chances for increasing healthy lifespan via optimization of late-life physiology.

While many studies claim to uncouple reduced reproduction from increased survival, we need inter-generational studies assessing Darwinian fitness in realistically complex and ecologically relevant environments because selection can favour different traits in different contexts. It is unlikely that ageing in any one taxon will be entirely driven by either energy or function trade-offs, not the least because some trade-offs may involve both. However, their relative

importance in the evolution of ageing across different taxa could certainly differ. For example, ageing in *C. elegans* might be driven primarily by suboptimal gene expression in adulthood, while ageing in mice by competitive energy allocation. At this point, we do not have the final answer even for these two extensively studied model organisms, let alone other animals. Until we know the answer across the broad range of taxonomic groups, ageing will remain an unsolved problem in biology.

Data accessibility. This article has no additional data.

Authors' contributions. A.A.M. and T.C. wrote the paper together.

Competing interests. We declare we have no competing interests.

Funding. This work has been supported by BBSRC (grant no. BB/R017387/1) and ERC Consolidator Grant GermlineAgeingSoma to A.A.M. and NERC (grant no. NE/R010056/1) to T.C.

Acknowledgements. We thank Locke Rowe (Toronto) and Martin Lind (Uppsala) for valuable discussions and comments, and Martin Lind for help with illustrations.

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
