## [Reviewer comments · Proceedings of the Royal Society B: Biological Sciences]

Review History

RSPB-2019-1604.R0 (Original submission)

Review form: Reviewer 1 (Joao Pedro de Magalhaes)

Recommendation

Accept with minor revision (please list in comments)

Scientific importance: Is the manuscript an original and important contribution to its field?

Excellent

General interest: Is the paper of sufficient general interest?

Good

Quality of the paper: Is the overall quality of the paper suitable?

Excellent

Is the length of the paper justified?

Yes

Should the paper be seen by a specialist statistical reviewer?

No

Do you have any concerns about statistical analyses in this paper? If so, please specify them explicitly in your report.

No

It is a condition of publication that authors make their supporting data, code and materials available - either as supplementary material or hosted in an external repository. Please rate, if applicable, the supporting data on the following criteria.

Is it accessible?

N/A

Is it clear?

N/A

Is it adequate?

N/A

Do you have any ethical concerns with this paper?

No

Comments to the Author

In this work, the authors review the evolutionary theory of aging in particular from the perspective of its physiological implications and trade-offs, with particular focus on the disposable soma theory and the developmental theory of aging. I think it is a timely and interesting work, and the manuscript is very well-written and makes for a compelling read.

I really only have two minor suggestions:

Page 10: In this discussion the authors could mention the naked mole rat in which queens can reproduce throughout their lives, often with higher temperature and metabolic demands (please see: <https://www.ncbi.nlm.nih.gov/pubmed/25832892>), and yet there is no evidence that they have a shorter lifespan.

Page 14, line 278, the authors could also cite another work in worms by the same authors that tested the role of developmental genes in aging:
<https://www.ncbi.nlm.nih.gov/pubmed/23144747>

Overall, I think this is an interesting and timely paper that will make a fine contribution to the literature.

It my usual policy to reveal my identity to the authors: Joao Pedro de Magalhaes.

Decision letter (RSPB-2019-1604.R0)

20-Aug-2019

Dear Alexei,

I apologise for the delay in getting a decision to you but, presumably because of the summer

holidays rather than the topic, it has been hard to get timely referees' reports. Anyway, I am pleased to inform you that your manuscript RSPB-2019-1604 entitled "Evolution of Ageing as a Tangle of Trade-Offs: Energy versus Function" has been accepted for publication in Proceedings B.

As I said, it was hard to get referees so, although we have one referee's report, I don't want to delay you any further. I feel I know enough about this topic to be confident that you have covered the expected material thoroughly and I agree with the referee, Joao Pedro de Magalhaes, who has signed his report, that it is very well written. Dr de Magalhães has recommended publication, but also suggest some minor revisions to your manuscript. Therefore, I invite you to respond to the comments and revise your manuscript. Because the schedule for publication is very tight, it is a condition of publication that you submit the revised version of your manuscript within 7 days. If you do not think you will be able to meet this date please let us know.

Best wishes,

Innes

Prof Innes Cuthill
Reviews Editor, Proceedings B
mailto: proceedingsb@royalsociety.org

Reviewer(s)' Comments to Author:

Referee: 1

Comments to the Author(s)

In this work, the authors review the evolutionary theory of aging in particular from the perspective of its physiological implications and trade-offs, with particular focus on the disposable soma theory and the developmental theory of aging. I think it is a timely and interesting work, and the manuscript is very well-written and makes for a compelling read.

I really only have two minor suggestions:

Page 10: In this discussion the authors could mention the naked mole rat in which queens can reproduce throughout their lives, often with higher temperature and metabolic demands (please see: <https://www.ncbi.nlm.nih.gov/pubmed/25832892>), and yet there is no evidence that they have a shorter lifespan.

Page 14, line 278, the authors could also cite another work in worms by the same authors that tested the role of developmental genes in aging:
<https://www.ncbi.nlm.nih.gov/pubmed/23144747>

Overall, I think this is an interesting and timely paper that will make a fine contribution to the literature.

It my usual policy to reveal my identity to the authors: Joao Pedro de Magalhaes.

Author's Response to Decision Letter for (RSPB-2019-1604.R0)

See Appendix A.

Decision letter (RSPB-2019-1604.R1)

27-Aug-2019

Dear Dr Maklakov

I am pleased to inform you that your manuscript entitled "Evolution of Ageing as a Tangle of Trade-Offs: Energy versus Function" has been accepted for publication in Proceedings B.

If you are likely to be away from e-mail contact during this period, let us know. Due to rapid publication and an extremely tight schedule, if comments are not received, we may publish the paper as it stands.

Open access

You are invited to opt for open access via our author pays publishing model. Payment of open access fees will enable your article to be made freely available via the Royal Society website as

soon as it is ready for publication. For more information about open access publishing please visit our website at http://royalsocietypublishing.org/site/authors/open_access.xhtml.

The open access fee is £1,700 per article (plus VAT for authors within the EU). If you wish to opt for open access then please let us know as soon as possible.

Paper charges

Sincerely,

Proceedings B
mailto: proceedingsb@royalsociety.org

Appendix A

Response to Reviewers

We were very happy to see that the Editor and Reviewer 1 (Joao Pedro de Magalhaes) were very positive about our review.

Following Reviewer's suggestion, we added another citation to the work from Curran's lab that looked at the post-development knock-downs in *C. elegans*.

We refrained from adding a citation to naked mole rate work on Page 10 because we felt it did not fit perfectly with the general argument that we were making there. We hope this is fine.

Kind regards,

Alex Maklakov and Tracey Chapman